# Learning from Linear Algebra: A Graph Neural Network Approach to Preconditioner Design for Conjugate Gradient Solvers

## Abstract

Large linear systems are ubiquitous in modern computational science and engineering. The main recipe for solving them is the use of Krylov subspace iterative methods with well-designed preconditioners. Deep learning models can be used as nonlinear preconditioners during the iteration of linear solvers such as the conjugate gradient (CG) method. Neural network models require an enormous number of parameters to approximate well in this setup. Another approach is to take advantage of small graph neural networks (GNNs) to construct preconditioners with predefined sparsity patterns. Recently, GNNs have been shown to be a promising tool for designing preconditioners to reduce the overall computational cost of iterative methods by constructing them more efficiently than with classical linear algebra techniques. However, preconditioners designed with these approaches cannot outperform those designed with classical methods in terms of the number of iterations in CG. In our work, we recall well-established preconditioners from linear algebra and use them as a starting point for training the GNN to obtain preconditioners that reduce the condition number of the system more significantly. Numerical experiments show that our approach outperforms both classical and neural network-based methods for an important class of parametric partial differential equations. We also provide a heuristic justification for the loss function used and show that preconditioners obtained by learning with this loss function reduce the condition number in a more desirable way for CG.

## 1 Introduction

Modern computational science and engineering problems are often based on parametric partial differential equations (PDEs). The lack of analytical solutions for realistic engineering problems (heat transfer, fluid flow, structural mechanics, etc.) leads researchers to exploit advances in numerical analysis. The basic numerical methods for solving PDEs, such as finite element, finite difference, finite volume and meshless methods (e.g., smoothed particle hydrodynamics), result in a system of linear equations $Ax = b$, $A \in \mathbb{R}^{n \times n}$, $x \in \mathbb{R}^n$, and $b \in \mathbb{R}^n$. These systems are usually sparse, i.e. the number of non-zero elements is $\ll n^2$. Furthermore, some classes of parametric PDEs are characterized by a very large dimension of the parametric space and by a high variation of the parameters for a given sample.

Typically, the application of parametric PDEs produces large linear systems, often with entries of varying scale, and therefore poses significant computational challenges. Projection Krylov subspace iterative methods are widely used to solve such systems. They rely on finding an optimal solution in a subspace constructed as follows: $\mathcal{K}_r(A, b) = \text{span}\{b, Ab, A^2b, \ldots, A^{r-1}\}..$

The conjugate gradient (CG) method is used to solve large sparse systems with symmetric and positive definite (SPD) matrices (Saad, 2003; Axelsson, 1996). CG has a well-established convergence theory and convergence guarantees for any symmetric matrix. However, the convergence rate of CG is determined by $\sqrt{\kappa(A)}$. The condition number $\kappa(A)$ of an SPD matrix $A$, defined in 2-norm, is a ratio between the maximum and minimum eigenvalues $\kappa(A) = |\lambda_{\max}|/|\lambda_{\min}|$.

Real-world applications with non-smooth high-contrast coefficient functions and high-dimensional linear systems separate eigenvalues and results into ill-conditioned problems. Decades of research in numerical linear algebra have been devoted to constructing preconditioners $P$ for ill-conditioned $A$ to improve the condition number in the form (for left-preconditioned systems) $\kappa(P^{-1}A) \ll \kappa(A)$.

The well-designed preconditioner should tend to approximate $A$, be easily invertible and be sparse or admit an efficient matrix-vector product. The construction of a preconditioner is typically a trade-off between the quality of the approximation and the cost of storage/inversion of the preconditioner (Saad, 2003).

In this paper, we propose a novel neural method for preconditioner design called PreCorrector (short for Preconditioner Corrector). Preconditioners constructed with PreCorrector have better effect on the spectrum than classical preconditioners by learning correction for the latter. Our contributions are as follows:

1. We propose a novel scheme for preconditioner design based on learning correction for well-established preconditioners from linear algebra with the GNN.

2. We propose an understanding of the loss function used with emphasis on low frequencies. We also provide experimental justification for the understanding of learning with such an objective.

3. We propose a novel dataset generation approach with a measurable complexity metric that addresses real-world problems.

4. We provide extensive experiments with varying matrix sizes and dataset complexities to demonstrate the superiority of the proposed approach and loss function over classical preconditioners.

## 2 NEURAL DESIGN OF PRECONDITIONER

**Related work**  While there are a dozen different preconditioners in linear algebra, for example (Saad, 2003; Axelsson, 1996): block Jacobi preconditioner, Gauss-Seidel preconditioner, sparse approximate inverse preconditioner, algebraic multigrid methods, etc., the choice of preconditioner depends on the specific problem, and practitioners often rely on a combination of theoretical understanding and numerical experimentation to select the most effective preconditioner. Even a brief description of all of them is beyond the scope of a single research paper. One can refer to the related literature for more details

The growing popularity of neural operators for learning mappings between infinite dimensional spaces (e.g., (Hao et al., 2024; Cao et al., 2024; Raonic et al., 2024)) is also present in recent work on using neural networks to speed up iterative solvers. The FCG-NO (Rudikov et al., 2024) approach combines neural operators with the conjugate gradient method to act as a nonlinear preconditioner for the flexible conjugate gradient method (Notay, 2000). This method uses a proven convergence bound as a training loss. A novel class of hybrid preconditioners (Kopaničáková & Karniadakis, 2024) combines DeepONet with standard iterative methods to solve parametric linear systems. This framework uses DeepONet for low-frequency error components and conventional methods for high-frequency components. The HINTS (Zhang et al., 2022) method integrates traditional relaxation techniques with DeepONet. It targets different spectral regions, ensuring a uniform convergence rate. It is also possible to use convolutional neural networks to speed up multigrid method (Azulay & Treister, 2022; Li et al., 2024), which require materialization of sparse matrices into dense format. However, these approaches can suffer from the curse of dimensionality when applied to large linear systems and can be too expensive to apply at each iteration step.

The authors of (Li et al., 2023; Häusner et al., 2023) present a novel approach to preconditioner design using GNNs that aim to approximate the matrix factorization and use it as a preconditioner. These approaches use shallow GNNs and typically require a single inference before the iteration process. GNNs take the initial left hand side matrix and right hand side vector as input and construct preconditioners in the form of a Cholesky decomposition. However, these GNNs cannot produce preconditioners that have a better effect on the condition number of the solving system than their classical analogues.

**Problem statement**   We consider systems of linear algebraic equations from the discretization of differential operators $Ax = b$ formed with a symmetric positive definite (SPD) matrix $A \succ 0$. One can use Gaussian elimination of complexity $\mathcal{O}(n^3)$ to solve small linear systems, but not real-world problems, which produce large and ill-conditioned systems.

**Preconditioned linear systems**   Before solving initial systems by iterative methods, we want to obtain a preconditioner $P$ such that the preconditioned linear system $P^{-1}Ax = P^{-1}b$ has a lower condition number than the initial system. If one knows the sparsity pattern of $A$, then possible options are incomplete LU decompositions (ILU) (Saad, 2003): (i) with $p$-level of fill-in denoted as ILU($p$) and (ii) ILU decomposition with threshold with $p$-level of fill-in denoted as ILUt($p$). Additional information about these preconditioners can be found in the Appendix A.7.

In this paper we focus on the SPD matrices so instead of ILU, ILU($p$) and ILUt($p$) we use the incomplete Choletsky factorization IC, IC($p$) and ICt($p$). Further, we will form the preconditioners in the form of Choletsky decomposition (Trefethen & Bau, 2022) $P = LL^\top$ with sparse $L$ obtained by different methods.

**Preconditioners with neural networks**   Our utlimate goal is to find such a decomposition that $\kappa((L(\theta)L(\theta)^\top)^{-1}A) \ll \kappa((LL^\top)^{-1}A) \ll \kappa(A)$, where $L$ is the classical numerical IC decomposition and $L(\theta) = \mathcal{F}(A)$ is an approximate decomposition with some function $\mathcal{F}$. Several papers (Li et al., 2023; Häusner et al., 2023) suggest using GNN as a function $\mathcal{F}$ to minimize certain loss function:

$$L(\theta) = \text{GNN}(\theta, A, b). \tag{1}$$

**Loss function**   The key question is which objective function to minimize in order to construct a preconditioner. A natural choice, which is also used in (Häusner et al., 2023), is:

$$\min \left\| P - A \right\|_F^2. \tag{2}$$

By design, this objective minimizes high frequency components (large eigenvalues), which is not desired. Low frequency components (small eigenvalues) are the most important because they correspond to the simulated phenomenon, when high frequency comes from discretization methods. It is also known that CG eliminates errors corresponding to high frequencies first and struggles the most with low frequencies. We suggest using $A^{-1}$ as the weight for the previous optimization objective to take into account low frequency since $\lambda(A) = \lambda^{-1}(A^{-1})$:

$$\min \left\| (P - A)A^{-1} \right\|_F^2 \tag{3}$$

Let us rewrite this objective using Hutchinson's estimator (Hutchinson, 1989):

$$\left\| (P - A)A^{-1} \right\|_F^2 = \left\| PA^{-1} - I \right\|_F^2 = \text{Tr}\left( (PA^{-1} - I)^\top (PA^{-1} - I) \right)$$
$$= \mathbb{E}_\varepsilon \left[ \varepsilon^\top (PA^{-1} - I)^\top (PA^{-1} - I)\varepsilon \right] = \mathbb{E}_\varepsilon \left\| (PA^{-1} - I)\varepsilon \right\|_2^2, \quad \varepsilon \sim \mathcal{N}(0,1). \tag{4}$$

Suppose we have a dataset of linear systems $A_i x_i = b_i$, then the training objective with $\varepsilon = b_i$, $P = L(\theta)L(\theta)^\top$ and $A_i^{-1}b_i = x_i$ will be:

$$\mathcal{L} = \frac{1}{N} \sum_{i=1}^N \left\| L(\theta)L(\theta)^\top x_i - b_i \right\|_2^2 \tag{5}$$

This loss function has appeared previously in related research (Li et al., 2023) but with an understanding of the inductive bias from the PDE data distribution. We claim that training with loss (3) allows to obtain better preconditioners than with loss (2). In the Section 5, we demonstrate that loss (3) does indeed mitigate low-frequency components.

## 3 LEARN CORRECTION FOR ILU

Our main goal is to construct preconditioners that reduce the condition number of an SPD matrix more than classical preconditioners with the same sparisy pattern.

### 3.1 GRAPH NEURAL NETWORK WITH PRESERVING SPARSITY PATTERN

Following the idea of (Li et al., 2023), we use the message-passing GNN architecture (Zhou et al., 2020) to preserve the sparsity pattern and predict the lower triangular matrix to create a preconditioner in a form of IC decomposition.

The duality between sparse matrices and graphs is used to obtain vertices and edges, such as $Ax = b \rightarrow \mathcal{G} = (\mathcal{V}, \mathcal{E})$, where $a_{i,j} = e_{i,j} \in \mathcal{E}, b_i = v_i \in \mathcal{V}$. The original GNN architecture from (Li et al., 2023):

1. First step is to use node and edge encoders to increase their dimensionality with multi-layer perceptrons (MLPs): $v_i = \text{MLP}_v(v_i), e_{i,j} = \text{MLP}_e(e_{i,j})$.

2. Then the encoded graph is processed with $T$ rounds of message passing (Brandstetter et al., 2022) $(t = 1, \ldots, T)$ to transfer information between vertices and edges. During a single round, we update vertices with $v_{i,t+1} = \text{MLP}_{mp,v}(v_{i,t}, \sum_j e_{i,j,t} v_{j,t})$, and then update the edges with $e_{i,j,t+1} = \text{MLP}_{mp,e}(e_{i,j,t}, v_{i,t+1} v_{j,t+1})$, for $i \neq j$.

3. Next step is to decode the lower triangular matrix while preserving the information in the upper triangular part of the matrix. To do this we average the bidirectional edges, decode them with MLP and then zero out the upper triangular part: $e_{i,j,T} = (e_{i,j,T} + e_{j,i,T})/2$ and $L_{i,j|i \leq j} = \text{MLP}_{\text{decod}}(e_{i,j,T}), L_{i,j|i>j} = 0$.

4. After all round of message passing the diagonal of the decomposition inherited as the diagonal from original matrix to ensure SPD property in resulting decomposition $\text{diag}(L(\theta)) := \sqrt{\text{diag}(A)}$.

5. Finally, assemble the preconditioner in a form of Choletsky decomposition $P := L(\theta)L(\theta)^\top$.

In our experiments, we observe that training GNN from scratch can be unstable and results in preconditioners that have weaker effect on the spectrum than their classical analogues. Moreover, with growing matrix size, the very first step of training, when GNN is initialized with random weights, overflows loss since the residual with random $P$ is huge. We propose to solve both problems by learning corrections to classical preconditioners.

### 3.2 PRECORRECTOR

Instead of passing left hand side matrix $A$ as input to GNN in (1), we propose: (i) to pass $L$ from the IC decomposition to the GNN and (ii) to train GNN to predict a correction for this decomposition (Figure 1):

$$L(\theta) = L + \alpha \cdot \text{GNN}(\theta, L, b). \tag{6}$$

The correction coefficient $\alpha$ is also a learning parameter that is updated during training. At the beginning of training, we set $\alpha = 0$ to ensure that the first gradient updates come from pure IC factorization. Since we already start with a good initial guess, we observed that pinning the diagonal is redundant and limits the training of the PreCorrector. Moreover, GNN in (6) takes as input the lower-triangular matrix $L$ from IC instead of $A$, so we are not anchored to a single specific sparse pattern of $A$ and we can: (i) omit half of the graph and speed up the training process and (ii) use different sparsity patterns to obtain even better preconditioners. In Experiment section, we show that the proposed approach with input $L$ from IC(0) and ICt(1) produces better preconditioners compared to classical IC(0) and ICt(1) and previous preconditioners using neural networks.

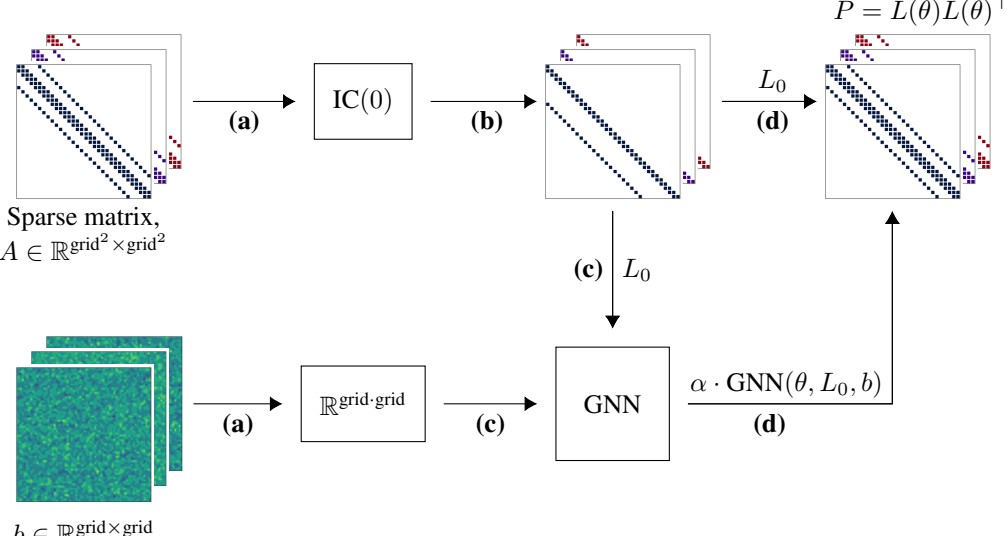

Figure 1: PreCorrector scheme that takes IC(0) as input. (a) Start with linear system $Ax = b$. (b) Obtain $L_0$ from IC(0) decomposition. (c) Input $L_0$ and $b$ to GNN. (d) Calculate $L(\theta)$ with equation 6 and construct preconditioner in form of IC. (**Very right picture**) Note that obtained preconditioner in form of IC(0) decomposition can be stored as initial matrix $A$.

## 4 DATASET

We test PreCorrector on SPD matrices obtained by discretization of elliptic equations. We consider a 2D diffusion equation:

$$
\begin{aligned}
-\nabla \cdot \big(k(x)\nabla u(x)\big) &= f(x), \text{ in } \Omega \\
u(x)\Big|_{x \in \partial \Omega} &= 0
\end{aligned}
, \tag{7}
$$

and 2D Poisson equation:

$$
\begin{aligned}
-\nabla^2 u(x) &= f(x), \text{ in } \Omega \\
u(x)\Big|_{x \in \partial \Omega} &= 0
\end{aligned}
, \tag{8}
$$

where $k(x)$ is a diffusion coefficient, $u(x)$ is a solution and $f(x)$ is a forcing term.

The diffusion equation is chosen because it occurs frequently in many engineering applications, such as: composite modeling (Carr & Turner, 2016), geophysical surveys (Oristaglio & Hohmann, 1984), fluid flow modeling (Muravleva et al., 2021). In these industrial applications, the coefficient functions are discontinuous, i.e. they change rapidly within neighbouring cells. Examples include the flow of immiscible fluids of different viscosities and fluid flow in heterogeneous porous media.

The condition number of a linear system depends on both the grid size and the contrast, but usually in scientific machine learning research, high contrast is not taken into account. In Section 5.1 we demonstrate that the previous approach can handle growing matrix size with constant coefficients quite well, but faces problems with growing contrast in the coefficients.

We propose to use a Gaussian Random Field (GRF) $\phi(x)$ to generate the coefficients. To control the complexity of the diffusion equation with discontinuous coefficients, we can measure the contrast in the GRF:

$$\text{contrast} = \exp \big( \max \big( \phi(x) \big) - \min \big( \phi(x) \big) \big). \tag{9}$$

Then we generate coefficients for the equation (7) as $k(x) = \exp(\phi(x))$.

By changing the variance in the GRF, we can control the contrast of the coefficients and thus the complexity. We generate datasets for each grid value from $\{32,\ 64,\ 128\}$. The contrast in the diffusion equation is controlled with a variance in the coefficient function GRF and takes value from $\{0.1,\ 0.5,\ 0.7\}$. The Gaussian covariance model is used in GRF. The forcing term $f$ is sampled from the standard normal distribution $\mathcal{N}(0,\ 1)$ and each PDE is discretized using the 5-point finite difference method. GRF is generated using the `parafields` library[1]. More details about datasets can be found in the Appendix A.6.

## 5 Experiments

In our approach, we used both IC(0) and ICt(1) to train the PreCorrector. In the next section, we will use the following notations:

- IC(0), ICt(1) and ICt(5) are classical preconditioners from linear algebra with a corresponding level of fill-in.
- PreCor$\big[$IC(0)$\big]$ and PreCor$\big[$ICt(1)$\big]$ is PreCorrector with corresponding preconditioners as input.

**Metrics** The main comparison of preconditioners designed with different algorithms are made by comparing total time including preconditioner construction time and the number of CG iterations to achieve a given tolerance. For construction time, we report averaged values over 200 runs of preconditioner construction and for CG time and iterations we report averaged values over the test set as well as standard deviations for the average values. Construction time for PreCorrector is reported including construction time of classical preconditioners.

The main idea behind using GNNs to construct preconditioners is to preserve the sparsity pattern. Therefore, the algorithmic complexity of using preconditioners (matrix-vector product) is the same when using preconditioners with the same sparsity pattern. This allows a fair evaluation of the quality of neural preconditioners with the same sparsity pattern only in terms of the number of CG iterations. Furthermore, all approaches to IC decomposition with the same sparsity pattern, including the classical ones, compete with each other in terms of construction time, effect on the spectrum (i.e., number of CG iterations) and generalization ability.

**Experiments environment** Each dataset in the Section 4 consists of 1000 training and 200 test linear systems. The final neural networks are trained with batch size 8, learning rate $10^{-3}$ and Adam optimizer. For a fair comparison, we set the GNN architecture to 5 message passing rounds and 2 hidden layers with 16 hidden features in all MLPs (see Section 3.1) in each experiment. PreCorrector training always starts with the parameter $\alpha = 0$ in (6). For GNN training, we used libraries from the JAX ecosystem: `jax` (Bradbury et al., 2018), `optax` (Kidger & Garcia, 2021), `equinox` (Deep-Mind et al., 2020). We used a single GPU Nvidia A40 48Gb for training. The construction time of preconditioners with neural design was measured on the same GPU. Preconditioners with classical algorithms were generated on the Intel(R) Xeon(R) Gold 6342 CPU @ 2.80GHz with `ilupp` library[2] Mayer (2007). The CG method was run on the same CPU using the `scipy` (Virtanen et al., 2020) implementation.

### 5.1 Preconditioners comparison

**Experiments with classical algorithms** Preconditioners constructed with PreCorrector outperform classical algorithms with the same sparsity pattern in both total time and effect on the spectrum, i.e., CG iterations (Table 1). While the IC(0) construction algorithm is simple and faster than the

---

[1]https://github.com/parafields/parafields
[2]https://github.com/c-f-h/ilupp

Table 1: Comparison on diffusion equation with variance 0.7: classical algorithms for IC(0) and ICt(1) and PreCorrector. Pre-time stands for precomputations time.

| Grid | Method | Pre-time | Time (iters) to $10^{-3}$ | Time (iters) to $10^{-6}$ | Time (iters) to $10^{-9}$ |
|---|---|---|---|---|---|
| $64 \times 64$ | IC(0) | $1.6 \cdot 10^{-4}$ | $0.169 \pm 0.006$ $(77 \pm 2.9)$ | $0.213 \pm 0.007$ $(98 \pm 3.2)$ | $0.248 \pm 0.007$ $(115 \pm 3.1)$ |
| | PreCor$\big[$IC(0)$\big]$ | $2.0 \cdot 10^{-3}$ | $\mathbf{0.112} \pm 0.097$ $(42 \pm 1.7)$ | $\mathbf{0.141} \pm 0.098$ $(55 \pm 2.1)$ | $\mathbf{0.170} \pm 0.099$ $(67 \pm 2.5)$ |
| | ICt(1) | $8.7 \cdot 10^{-4}$ | $0.105 \pm 0.004$ $(46 \pm 1.8)$ | $0.133 \pm 0.004$ $(59 \pm 1.9)$ | $0.155 \pm 0.004$ $(69 \pm 1.9)$ |
| | PreCor$\big[$ICt(1)$\big]$ | $2.3 \cdot 10^{-3}$ | $\mathbf{0.091} \pm 0.070$ $(29 \pm 1.4)$ | $\mathbf{0.115} \pm 0.071$ $(38 \pm 1.7)$ | $\mathbf{0.138} \pm 0.072$ $(47 \pm 1.9)$ |
| $128 \times 128$ | IC(0) | $5.1 \cdot 10^{-4}$ | $1.071 \pm 0.189$ $(156 \pm 5.6)$ | $1.338 \pm 0.229$ $(196 \pm 5.5)$ | $1.554 \pm 0.261$ $(228 \pm 5.7)$ |
| | PreCor$\big[$IC(0)$\big]$ | $2.0 \cdot 10^{-3}$ | $\mathbf{0.571} \pm 0.078$ $(67 \pm 3.2)$ | $\mathbf{0.720} \pm 0.079$ $(85 \pm 3.5)$ | $\mathbf{0.859} \pm 0.079$ $(102 \pm 3.9)$ |
| | ICt(1) | $3.8 \cdot 10^{-3}$ | $0.720 \pm 0.098$ $(95 \pm 3.5)$ | $0.902 \pm 0.122$ $(119 \pm 3.4)$ | $1.048 \pm 0.141$ $(139 \pm 3.5)$ |
| | PreCor$\big[$ICt(1)$\big]$ | $5.4 \cdot 10^{-3}$ | $\mathbf{0.470} \pm 0.071$ $(50 \pm 2.5)$ | $\mathbf{0.593} \pm 0.078$ $(64 \pm 2.9)$ | $\mathbf{0.708} \pm 0.086$ $(77 \pm 3.1)$ |

Table 2: Comparison on diffusion equation with variance 0.7: classical algorithms for ICt(5) and PreCor$\big[$ICt(1)$\big]$. Pre-time stands for precomputations time.

| Grid | Method | Pre-time | Time (iters) to $10^{-3}$ | Time (iters) to $10^{-6}$ | Time (iters) to $10^{-9}$ |
|---|---|---|---|---|---|
| $64 \times 64$ | ICt(5) | $2.1 \cdot 10^{-3}$ | $\mathbf{0.069} \pm 0.003$ $(22 \pm 0.8)$ | $\mathbf{0.087} \pm 0.003$ $(28 \pm 0.8)$ | $\mathbf{0.103} \pm 0.003$ $(34 \pm 0.8)$ |
| | PreCor$\big[$ICt(1)$\big]$ | $2.3 \cdot 10^{-3}$ | $0.091 \pm 0.070$ $(29 \pm 1.4)$ | $0.115 \pm 0.071$ $(38 \pm 1.7)$ | $0.138 \pm 0.072$ $(47 \pm 1.9)$ |
| $128 \times 128$ | ICt(5) | $8.9 \cdot 10^{-3}$ | $0.570 \pm 0.071$ $(48 \pm 1.7)$ | $0.705 \pm 0.088$ $(60 \pm 1.6)$ | $0.833 \pm 0.104$ $(70 \pm 1.8)$ |
| | PreCor$\big[$ICt(1)$\big]$ | $5.4 \cdot 10^{-3}$ | $\mathbf{0.470} \pm 0.071$ $(50 \pm 2.5)$ | $\mathbf{0.593} \pm 0.078$ $(64 \pm 2.9)$ | $\mathbf{0.708} \pm 0.086$ $(77 \pm 3.1)$ |

PreCor$\big[$IC(0)$\big]$ inference, the PreCor$\big[$IC(0)$\big]$ requires fewer iterations to achieve the required toler-ance. The more efficient preconditioner ICt(1) with a more complex construction algorithm makes up the difference with construction time of PreCor$\big[$ICt(1)$\big]$. The proposed approach based on this preconditioner, PreCor$\big[$ICt(1)$\big]$, has a better effect on spectrum of the initial $A$ than classical ICt(1).

In Table 2 one can observe that PreCor$\big[$ICt(1)$\big]$ generally performs on par with ICt(5) in terms of total time, which is denser than the initial left hand side $A$ (see Appendix A.6). While ICt(5) has fewer CG iterations, denser preconditioners have more operations when used during iterations. Consequently, for larger grid sizes, the total time of PreCor$\big[$ICt(1)$\big]$ is less than that of ICt(5). Moreover, the construction cost of ICt(5) scales worse than the inference of PreCorrector. Therefore, PreCorrector is a more farvorable method for large systems than effective dense preconditioners. Additional information on the scalability of the PreCorrector can be found in the Appendix A.5. One can find more results on different datasets in Appendix A.1 and Appendix A.2.

In our experiments, the value of the correction coefficient $\alpha$ is always negative when training with loss (3). Interestingly, when we train PreCorrector with loss (2), $\alpha$ is always positive. Detailed values of $\alpha$ can be found in the Appendix A.8.

Finally, certain engineering problems require iterative solvers to be run multiple times on the same matrices. For these problems, the preconditioner's precomputation overhead contributes less to the total time. We also observe a good generalization of our approach when transferring our preconditioner between grids and datasets (see Appendix A.4). The transferability of the PreCorrector allows both to use the trained PreCorrector on similar problems and to train the PreCorrector on the easier problems with inference on the hard ones.

Table 3: Comparison on Poisson equation: classical algorithms for IC(0), GNN from (Li et al., 2023) and PreCorrector. Pre-time stands for precomputations time.

| Grid | Method | Pre-time | Time (iters) to $10^{-3}$ | Time (iters) to $10^{-6}$ | Time (iters) to $10^{-9}$ |
|---|---|---|---|---|---|
| | IC(0) | $1.9 \cdot 10^{-4}$ | $0.134 \pm 0.001$ $(62 \pm 0.0)$ | $0.166 \pm 0.001$ $(77 \pm 0.0)$ | $0.199 \pm 0.001$ $(93 \pm 0.0)$ |
| $64 \times 64$ | PreCor$[$IC(0)$]$ | $2.0 \cdot 10^{-3}$ | $\mathbf{0.093} \pm 0.072$ $(31 \pm 0.3)$ | $\mathbf{0.118} \pm 0.072$ $(41 \pm 0.4)$ | $\mathbf{0.142} \pm 0.072$ $(50 \pm 0.1)$ |
| | (Li et al., 2023) | $2.5 \cdot 10^{-3}$ | $\underline{0.106} \pm 0.076$ $(33 \pm 0.0)$ | $\underline{0.131} \pm 0.077$ $(42 \pm 0.0)$ | $\underline{0.154} \pm 0.077$ $(50 \pm 0.0)$ |
| | IC(0) | $4.6 \cdot 10^{-4}$ | $0.901 \pm 0.116$ $(115 \pm 0.0)$ | $1.223 \pm 0.156$ $(157 \pm 0.5)$ | $1.412 \pm 0.181$ $(181 \pm 0.0)$ |
| $128 \times 128$ | PreCor$[$IC(0)$]$ | $2.0 \cdot 10^{-3}$ | $\mathbf{0.391} \pm 0.085$ $(48 \pm 0.7)$ | $\mathbf{0.497} \pm 0.093$ $(61 \pm 0.2)$ | $\mathbf{0.594} \pm 0.100$ $(73 \pm 0.5)$ |
| | (Li et al., 2023) | $2.5 \cdot 10^{-3}$ | $\underline{0.430} \pm 0.078$ $(50 \pm 0.0)$ | $\underline{0.537} \pm 0.079$ $(63 \pm 0.0)$ | $\underline{0.645} \pm 0.081$ $(76 \pm 0.0)$ |

Table 4: Comparison on diffusion equation with variance 0.1: classical algorithms for IC(0), GNN from (Li et al., 2023) and PreCorrector. 1) None of the test linear systems converged to $10^{-3}$ tolerance in 300 iterations.

| Grid | Method | Pre-time | Time (iters) to $10^{-3}$ | Time (iters) to $10^{-6}$ | Time (iters) to $10^{-9}$ |
|---|---|---|---|---|---|
| | IC(0) | $7.2 \cdot 10^{-5}$ | $\underline{0.043} \pm 0.001$ $(32 \pm 0.6)$ | $\underline{0.055} \pm 0.002$ $(42 \pm 0.7)$ | $\underline{0.066} \pm 0.002$ $(50 \pm 0.6)$ |
| $32 \times 32$ | PreCor$[$IC(0)$]$ | $1.6 \cdot 10^{-3}$ | $\mathbf{0.040} \pm 0.092$ $(21 \pm 0.4)$ | $\mathbf{0.050} \pm 0.092$ $(28 \pm 0.4)$ | $\mathbf{0.060} \pm 0.092$ $(35 \pm 0.4)$ |
| | (Li et al., 2023) | $2.5 \cdot 10^{-3}$ | $0.088 \pm 0.070$ $(60 \pm 5.9)$ | $0.113 \pm 0.071$ $(79 \pm 7.6)$ | $0.135 \pm 0.071$ $(96 \pm 9.1)$ |
| | IC(0) | $1.6 \cdot 10^{-4}$ | $0.145 \pm 0.010$ $(65 \pm 1.4)$ | $0.185 \pm 0.012$ $(84 \pm 1.0)$ | $0.218 \pm 0.014$ $(99 \pm 0.9)$ |
| $64 \times 64$ | PreCor$[$IC(0)$]$ | $2.0 \cdot 10^{-3}$ | $\mathbf{0.099} \pm 0.087$ $(33 \pm 0.6)$ | $\mathbf{0.126} \pm 0.088$ $(43 \pm 0.6)$ | $\mathbf{0.151} \pm 0.089$ $(53 \pm 0.6)$ |
| | (Li et al., 2023) | $2.5 \cdot 10^{-3}$ | nan[1] | nan[1] | nan[1] |

**Experiments with neural preconditioner design** Preconditioners constructed with GNNs from previous works (Li et al., 2023; Häusner et al., 2023) report speedup compared to classical preconditioners when the latter had a long construction time. At the same time, they cannot outperform the effect of IC(0) on the spectrum of the initial matrix. We implemented the approach of (Li et al., 2023) and compared it with PreCorrector and IC(0).

Experiments on simple Poisson equation show that both PreCor$[$IC(0)$]$ and GNN from (Li et al., 2023) have similar number of CG iterations. However, PreCor$[$IC(0)$]$ has lighter inference and thus less total time-to-solution. Note that constructing IC(0) with the `ilupp` library takes negligible

time comapred to the total CG time. With PDE coefficients generated as Gaussian random fields with small discontinuities in them, the GNN from (Li et al., 2023) cannot reach the same number of CG iterations as the classical IC(0), while quality of the PreCorrector does not degenerate when it is trained on the datasets with higher contrast.

Results for GNN from (Li et al., 2023) are obtained with the same GNN hyperparameters and architecture as reported in the original paper with only one change: GNN is initialized with all weights and biases equal to zero and trained untill convergence. Without this change in architecture, the training did not converge. We also observed unstable training of the GNN from (Li et al., 2023) and unpredictable quality of the preconditioner depending on the training length. In our experiments, we could get a better preconditioner if we did not train until convergence. However, we could not predict the time for early stopping except by directly computing $\kappa(P^{-1}A)$, which is too expensive.

## 5.2 Loss function

The equivalence of the losses (3) and (5) allows to avoid explicit inverse materialization and provides maximum complexity of the matrix-vector product in the loss during training. Recall that $A$ comes from the 5-point finite difference discretization of the diffusion equation (Section 4). $A$ tends to a diagonal matrix with $n \to \infty$ and we can assume that $A$ is a diagonal matrix for sufficiently large linear systems. Minimizing a matrix product between the preconditioner and $A^{-1}$ in (3) makes the eigenvalues tend to 1.

As mentioned in Section 2, one should focus on approximating the low frequency components. The loss (3) does indeed reduce the distance between the extreme eigenvalues compared to IC(0) which can be observed in Table 5. Moreover, the gap between the extreme eigenvalues is covered by the increase in the minimum eigenvalue, which supports the hypothesis of low frequency cancellation. The maximum eigenvalue also increase, but by a much smaller order of magnitude.

In Table 5 we have also included recently obtained bounds for the minimum and maximum eigenvalues (Häusner et al., 2024):

$$
\begin{aligned}
&\lambda_{\min} \geq \|PA^{-1}\|_F^{-1} \geq \left(\|PA^{-1} - I\|_F + 1\right)^{-1} \\
&\lambda_{\max} \leq \|A - P\|_2 \|P^{-1}\|_2 + 1
\end{aligned} \tag{10}
$$

At the same time, the preconditioner trained with the loss function (2) without weighting with $A^{-1}$ gives a worse effect on spectrum of $A$ (Table 6) and takes much more time to converge (Figure 2). Different distributions of the eigenvalues can be found in Figure 3 in the Appendix A.3

Table 5: Condition number, spectrum and value of loss (3) for a sampled model from on diffusion equation with variance 0.7 on grid $128 \times 128$. The loss value is calculated directly with $A^{-1}$.

| Matrix | $\kappa(P^{-1}A)$ | $\lambda_{\min}$ | $\lambda_{\max}$ | $\left\|LL^\top A^{-1} - I\right\|_F^2$ | Bound $\lambda_{\min}$ | Bound $\lambda_{\max}$ |
|---:|---:|---:|---:|---:|---:|---:|
| $A$ | 87565 | 17.5506 | 1536813.4 | — | — | — |
| $\left(L_0 L_0^\top\right)^{-1} A$ | 749 | 0.0016 | 1.2 | $1.04 \cdot 10^6$ | $9.7 \cdot 10^{-4}$ | 230 |
| $\left(L(\theta)L(\theta)^\top\right)^{-1} A$ | 78 | 0.1981 | 15.5 | $1.73 \cdot 10^3$ | $8.4 \cdot 10^{-3}$ | 4157 |

## 6 Discussion

In our work, we propose a neural design of preconditioners for the CG iterative method that can outperform analogous classical preconditioners with the same sparsity pattern of the ILU family in terms of both effect on the spectrum and total time-to-solution. Using the classical preconditioners as a starting point and learning corrections for them, we achieve stable and fast training convergence that can handle parametric PDEs with contrast coefficients. We also propose a complexity metric to measure the complexity of PDEs with random coefficients.

Table 6: Comparison of losses (2) and (3). Number of CG iterations on the diffusion equation with 0.7 on grid $64 \times 64$. During training Hutchinson trick is applied for both losses. [1]Condition number and eigenvalues are calculated on a single sampled linear system.

| Loss | $10^{-3}$ | $10^{-6}$ | $10^{-9}$ | $\kappa(P^{-1}A)^1$ | $\lambda^1_{\min}$ | $\lambda^1_{\max}$ |
|---|---|---|---|---|---|---|
| (2) | $123 \pm 4.7$ | $155 \pm 5.3$ | $182 \pm 5.4$ | 571 | 0.0033 | **1.88** |
| (3) **(Ours)** | **$42 \pm 1.7$** | **$55 \pm 2.1$** | **$67 \pm 2.5$** | **60** | **0.1406** | 8.40 |

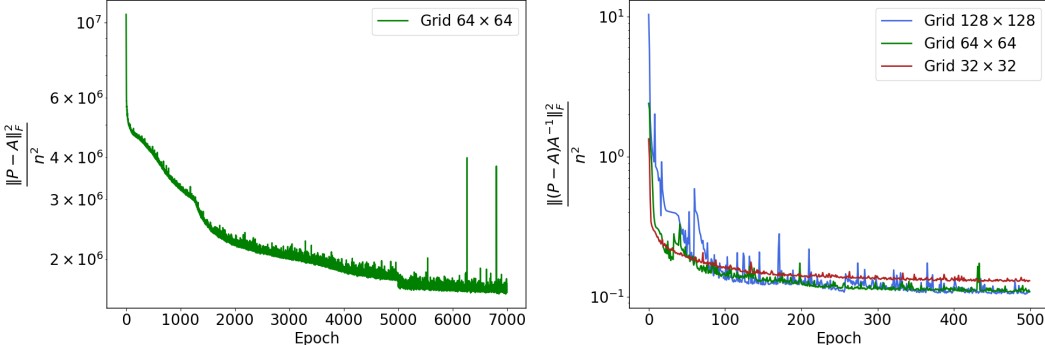

Figure 2: Test losses during training of $\mathrm{PreCor}\big[\mathrm{IC}(0)\big]$ on the diffusion equation with variance 0.7 During training Hutchinson trick is applied for both losses.

We provide numerical evidence for our observation of low-frequency cancellation with the loss function used. However, we found no trace of this relationship in the numerical analysis literature. We believe that there exists a learnable transformation that will be universal for different sparse matrices to construct the ILU decomposition that will significantly reduce $\kappa(A)$. We propose that this loss analysis is the key ingredient for successful learning of the general form transformation.

## 7 LIMITATIONS

The limitations of the proposed approach are as follows:

1. Theoretical study of the loss function used. We provide only a heuristic understanding with experimental justification for the loss function. A theoretical analysis of the loss function is the subject of future research.

2. The target objective in norms other than $\|\cdot\|_F$ may provide a tighter bound on the spectrum. Investigating the possible use of target values in other norms is a logical next step.

3. Experiments on other meshes and sparsity patterns of the resulting left hand side matrices $A$. Generalization of the PreCorrector to transformation in the space of sparse matrices with general sparsity patterns.

4. While the PreCorrector has only been tested on systems with SPD matrices from the discretization of elliptic equations, further work will require generalization to irregular grids, non-symmetric problems, hyperbolic PDEs, nonlinear problems, to other iterative solvers such as GMRES and BiCGSTAB and modification of the preconditioner design accordingly.

5. The forcing term $f(x)$ is sampled from the standard normal distribution, but the case of complex forcing terms needs to be studied separately as it can also affect the complexity of solving parametric PDEs.

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

# A APPENDIX

## A.1 ADDITIONAL EXPERIMENTS WITH IC(0) AND ICt(1) PRECONDITIONERS

Table 7: Comparison on diffusion equation with variance 0.1: classical algorithms for IC(0) and ICt(1) and PreCorrector. Pre-time stands for precomputations time.

| Grid | Method | Pre-time | Time (iters) to $10^{-3}$ | Time (iters) to $10^{-6}$ | Time (iters) to $10^{-9}$ |
|---|---|---|---|---|---|
| $64 \times 64$ | IC(0) | $1.6 \cdot 10^{-4}$ | $0.145 \pm 0.010$ $(65 \pm 1.4)$ | $0.185 \pm 0.012$ $(84 \pm 1.0)$ | $0.218 \pm 0.014$ $(99 \pm 0.9)$ |
| | PreCor$\big[$IC(0)$\big]$ | $2.0 \cdot 10^{-3}$ | $\mathbf{0.099} \pm 0.087$ $(33 \pm 0.6)$ | $\mathbf{0.126} \pm 0.088$ $(43 \pm 0.6)$ | $\mathbf{0.151} \pm 0.089$ $(53 \pm 0.6)$ |
| | ICt(1) | $8.6 \cdot 10^{-4}$ | $0.090 \pm 0.0020$ $(40 \pm 0.9)$ | $0.115 \pm 0.002$ $(51 \pm 0.8)$ | $0.136 \pm 0.001$ $(61 \pm 0.6)$ |
| | PreCor$\big[$ICt(1)$\big]$ | $2.3 \cdot 10^{-3}$ | $\mathbf{0.088} \pm 0.094$ $(23 \pm 0.5)$ | $\mathbf{0.112} \pm 0.094$ $(31 \pm 0.4)$ | $\mathbf{0.134} \pm 0.094$ $(38 \pm 0.4)$ |
| $128 \times 128$ | IC(0) | $4.6 \cdot 10^{-4}$ | $0.992 \pm 0.153$ $(131 \pm 3.2)$ | $1.267 \pm 0.191$ $(168 \pm 1.9)$ | $1.489 \pm 0.221$ $(198 \pm 1.9)$ |
| | PreCor$\big[$IC(0)$\big]$ | $2.0 \cdot 10^{-3}$ | $\mathbf{0.425} \pm 0.080$ $(52 \pm 1.5)$ | $\mathbf{0.538} \pm 0.086$ $(66 \pm 1.3)$ | $\mathbf{0.643} \pm 0.094$ $(80 \pm 1.6)$ |
| | ICt(1) | $3.4 \cdot 10^{-3}$ | $0.555 \pm 0.074$ $(80 \pm 1.9)$ | $0.709 \pm 0.092$ $(102 \pm 1.3)$ | $0.833 \pm 0.109$ $(120 \pm 1.1)$ |
| | PreCor$\big[$ICt(1)$\big]$ | $5.0 \cdot 10^{-3}$ | $\mathbf{0.367} \pm 0.069$ $(37 \pm 0.9)$ | $\mathbf{0.469} \pm 0.071$ $(48 \pm 0.8)$ | $\mathbf{0.560} \pm 0.074$ $(58 \pm 0.8)$ |

Table 8: Comparison on diffusion equation with variance 0.5: classical algorithms for IC(0) and ICt(1) and PreCorrector. Pre-time stands for precomputations time.

| Grid | Method | Pre-time | Time (iters) to $10^{-3}$ | Time (iters) to $10^{-6}$ | Time (iters) to $10^{-9}$ |
|---|---|---|---|---|---|
| $64 \times 64$ | IC(0) | $1.6 \cdot 10^{-4}$ | $0.165 \pm 0.067$ $(73 \pm 2.1)$ | $0.207 \pm 0.069$ $(93 \pm 2.1)$ | $0.242 \pm 0.070$ $(109 \pm 2.2)$ |
| | PreCor$\big[$IC(0)$\big]$ | $2.0 \cdot 10^{-3}$ | $\mathbf{0.098} \pm 0.095$ $(38 \pm 1.1)$ | $\mathbf{0.124} \pm 0.096$ $(50 \pm 1.3)$ | $\mathbf{0.148} \pm 0.097$ $(60 \pm 1.4)$ |
| | ICt(1) | $9.6 \cdot 10^{-4}$ | $0.101 \pm 0.003$ $(44 \pm 1.3)$ | $0.128 \pm 0.003$ $(56 \pm 1.3)$ | $0.150 \pm 0.003$ $(66 \pm 1.4)$ |
| | PreCor$\big[$ICt(1)$\big]$ | $2.4 \cdot 10^{-3}$ | $\mathbf{0.085} \pm 0.077$ $(27 \pm 0.8)$ | $\mathbf{0.107} \pm 0.077$ $(35 \pm 0.9)$ | $\mathbf{0.128} \pm 0.078$ $(43 \pm 1.0)$ |
| $128 \times 128$ | IC(0) | $4.7 \cdot 10^{-4}$ | $1.086 \pm 0.163$ $(147 \pm 4.3)$ | $1.367 \pm 0.203$ $(185 \pm 4.0)$ | $1.601 \pm 0.237$ $(217 \pm 4.0)$ |
| | PreCor$\big[$IC(0)$\big]$ | $2.0 \cdot 10^{-3}$ | $\mathbf{0.498} \pm 0.113$ $(62 \pm 2.7)$ | $\mathbf{0.628} \pm 0.123$ $(78 \pm 2.9)$ | $\mathbf{0.748} \pm 0.134$ $(94 \pm 3.4)$ |
| | ICt(1) | $3.4 \cdot 10^{-3}$ | $0.629 \pm 0.084$ $(89 \pm 2.6)$ | $0.791 \pm 0.107$ $(112 \pm 2.5)$ | $0.927 \pm 0.125$ $(132 \pm 2.5)$ |
| | PreCor$\big[$ICt(1)$\big]$ | $5.0 \cdot 10^{-3}$ | $\mathbf{0.344} \pm 0.083$ $(43 \pm 1.5)$ | $\mathbf{0.437} \pm 0.093$ $(55 \pm 1.8)$ | $\mathbf{0.525} \pm 0.101$ $(66 \pm 1.8)$ |

## A.2 ADDITIONAL EXPERIMENTS WITH ICT(5) PRECONDITIONER

Table 9: Comparison on diffusion equation with variance 0.1: classical algorithms for ICt(5) and PreCor$\big[$ICt(1)$\big]$. Pre-time stands for precomputations time.

| Grid | Method | Pre-time | Time (iters) to $10^{-3}$ | Time (iters) to $10^{-6}$ | Time (iters) to $10^{-9}$ |
|---|---|---|---|---|---|
| $64 \times 64$ | ICt(5) | $1.9 \cdot 10^{-3}$ | $\mathbf{0.070} \pm 0.003$ $(20 \pm 0.6)$ | $\mathbf{0.083} \pm 0.003$ $(27 \pm 0.4)$ | $\mathbf{0.098} \pm 0.003$ $(32 \pm 0.4)$ |
| | PreCor$\big[$ICt(1)$\big]$ | $2.3 \cdot 10^{-3}$ | $0.088 \pm 0.094$ $(23 \pm 0.5)$ | $0.112 \pm 0.094$ $(31 \pm 0.4)$ | $0.134 \pm 0.094$ $(38 \pm 0.4)$ |
| $128 \times 128$ | ICt(5) | $7.9 \cdot 10^{-3}$ | $0.488 \pm 0.074$ $(42 \pm 1.1)$ | $0.625 \pm 0.095$ $(54 \pm 0.8)$ | $0.736 \pm 0.112$ $(64 \pm 0.7)$ |
| | PreCor$\big[$ICt(1)$\big]$ | $5.0 \cdot 10^{-3}$ | $\mathbf{0.367} \pm 0.069$ $(37 \pm 0.9)$ | $\mathbf{0.469} \pm 0.071$ $(48 \pm 0.8)$ | $\mathbf{0.560} \pm 0.074$ $(58 \pm 0.8)$ |

Table 10: Comparison on diffusion equation with variance 0.5: classical algorithms for ICt(5) and PreCor$\big[$ICt(1)$\big]$. Pre-time stands for precomputations time.

| Grid | Method | Pre-time | Time (iters) to $10^{-3}$ | Time (iters) to $10^{-6}$ | Time (iters) to $10^{-9}$ |
|---|---|---|---|---|---|
| $64 \times 64$ | ICt(5) | $2.2 \cdot 10^{-3}$ | $\mathbf{0.068} \pm 0.002$ $(21 \pm 0.8)$ | $\mathbf{0.086} \pm 0.002$ $(28 \pm 0.7)$ | $\mathbf{0.112} \pm 0.002$ $(33 \pm 0.7)$ |
| | PreCor$\big[$ICt(1)$\big]$ | $2.4 \cdot 10^{-3}$ | $0.085 \pm 0.077$ $(27 \pm 0.8)$ | $0.107 \pm 0.077$ $(35 \pm 0.9)$ | $0.128 \pm 0.078$ $(43 \pm 1.0)$ |
| $128 \times 128$ | ICt(5) | $8.1 \cdot 10^{-3}$ | $0.465 \pm 0.050$ $(45 \pm 1.4)$ | $0.588 \pm 0.062$ $(57 \pm 1.2)$ | $0.689 \pm 0.073$ $(68 \pm 1.2)$ |
| | PreCor$\big[$ICt(1)$\big]$ | $5.0 \cdot 10^{-3}$ | $\mathbf{0.344} \pm 0.083$ $(43 \pm 1.5)$ | $\mathbf{0.437} \pm 0.093$ $(55 \pm 1.8)$ | $\mathbf{0.525} \pm 0.101$ $(66 \pm 1.8)$ |

## A.3 DISTRIBUTION OF EIGENVALUES

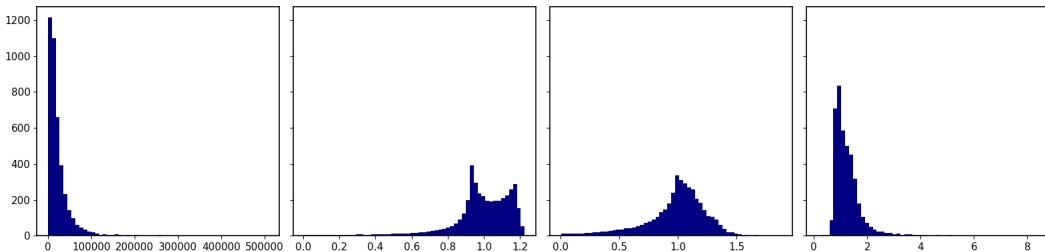

Figure 3: Distribution of eigenvalues for a sampled linear system of diffusion equation with variance 0.7 on grid $64 \times 64$. (**From left to right**): initial left hand side $A$, $A$ preconditioned with IC(0), $A$ preconditioned with PreCor$\big[$IC(0)$\big]$ trained with loss (2), $A$ preconditioned with PreCor$\big[$IC(0)$\big]$ trained with loss (3). Hutchinson estimator is used for both losses.

## A.4 GENERALIZATION

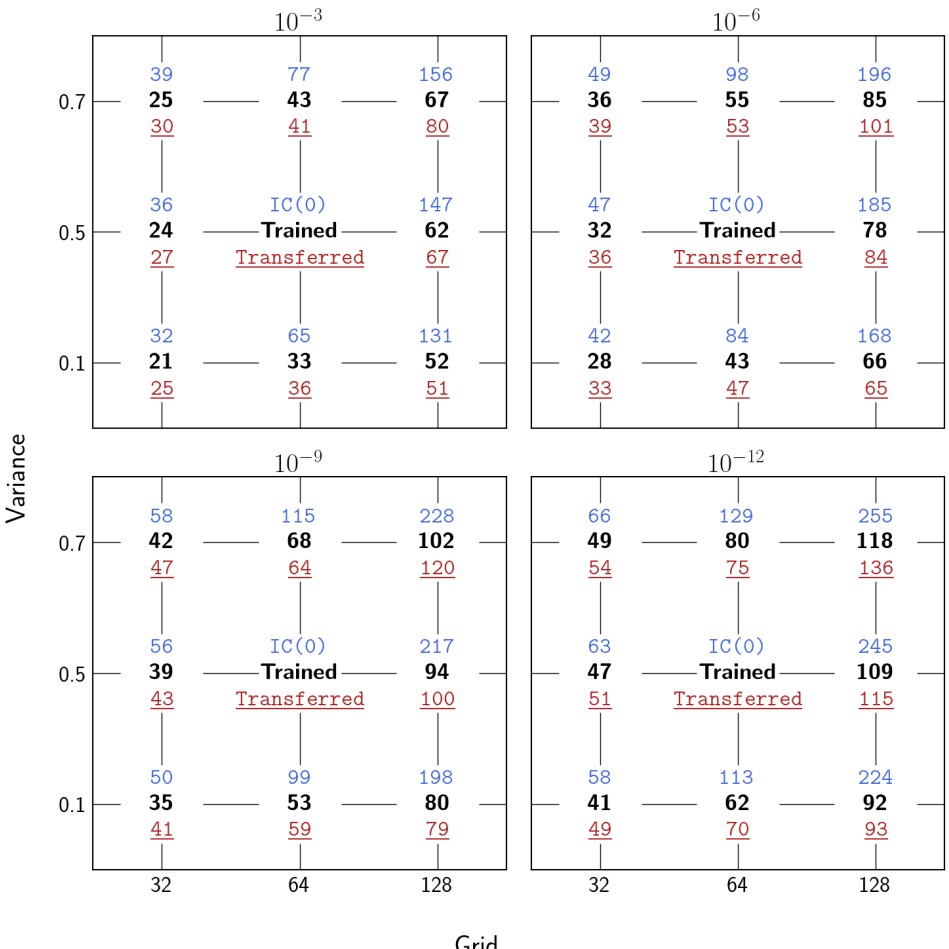

Figure 4: Generalization of PreCorrector on unseen datasets. Values are number of CG iterations to achieve required tolerance, which specified in the title of each plot. **Blue** – IC(0) is used as preconditioner. **Black** – PreCorrector$\big[$IC(0)$\big]$; trained and inferenced on the same dataset. **Red** – PreCorrector$\big[$IC(0)$\big]$; trained on the diffusion eqaution with variance $0.5$ on grid $64 \times 64$; inferenced on the dataset, that is described by axes values.

## A.5 SCALABILITY

The PreCorrector is a combination of several MLPs with message-passing connections. When growing non-zero elements nnz (i.e., growing a matrix size) with the same sparsity pattern, the complexity of the PreCorrector forward call grows linearly. The algorithm for IC(0) has $\mathcal{O}(\text{nnz})$. We expect a small increase in computational cost for larger systems, since with matrix grows we are only interested in the growth of non-zero elements in the matrix. The growth of non-zero elements is much less severe than the growth of a matrix size due to the nature of matrices – discretization of PDEs. The PreCorrector works directly on the edges and nodes of the graph and can be easily data-parallelized.

## A.6 Dataset description

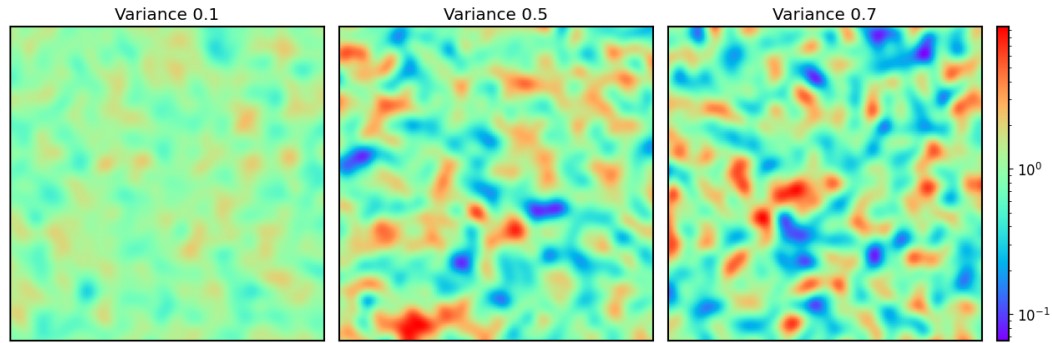

Figure 5: Coefficient function $k(x) = \exp\left(\phi(x)\right)$ for grid $128 \times 128$ with different variances.

Table 11: Contrast values for diffusion equation with various variances.

| Variance | Grid | Min contrast | Mean contrast | Max contrast |
|---|---|---|---|---|
| | 32 | 5 | 7 | 11 |
| 0.1 | 64 | 5 | 8 | 12 |
| | 128 | 6 | 8 | 14 |
| | 32 | 36 | 86 | 179 |
| 0.5 | 64 | 45 | 103 | 200 |
| | 128 | 50 | 116 | 297 |
| | 32 | 180 | 277 | 697 |
| 0.7 | 64 | 200 | 318 | 742 |
| | 128 | 300 | 426 | 798 |

Table 12: Size of linear systems and number of nonzero elements (nnz) for different grid sizes and matrices.

| Matrix | Grid $32 \times 32$ Size | Grid $32 \times 32$ nnz, % | Grid $64 \times 64$ Size | Grid $64 \times 64$ nnz, % | Grid $128 \times 128$ Size | Grid $128 \times 128$ nnz, % |
|---|---|---|---|---|---|---|
| $A$ | | 0.4761 | | 0.1205 | | 0.0303 |
| $L$ from IC(0) | 1024 | 0.2869 | 4096 | 0.0725 | 16384 | 0.0182 |
| $L$ from ICt(1) | | 0.3785 | | 0.0961 | | 0.0242 |
| $L$ from ICt(5) | | 0.7547 | | 0.1920 | | 0.0485 |

## A.7 Incomplete LU factorization

Full LU decomposition (Trefethen & Bau, 2022) for a square non-singular matrix defined as the product of the lower and upper triangular matrices $B = L_B U_B$. In general, these matrices have no restriction on the position and number of elements within their triangular structure and can even be dense for a sparse matrix $B$. On the other hand, an ILU is an approximate LU factorization:

$$A \approx LU, \tag{11}$$

where $LU - A$ satisfies certain constraints.

Zero fill-in ILU, denoted ILU(0), is an approximate LU factorization $A \approx L_0 U_0$ in such a way that $L_0$ has exactly the same sparsity pattern as the lower part of $A$ and $U_0$ has exactly the same sparsity pattern as the upper part of $A$. For the ILU($p$) decomposition the level of fill-in is defined hierarchically. The product of the factors of the ILU(0) decomposition produces a new matrix $B$ with a larger number of non-zero elements. The factors of the ILU(1) factorization have the same sparsity patterns as lower and upper parts of the sparsity pattern of the matrix $B$. With this recursion one gets a pattern of the ILU($p$) factorization with $p$-level of fill-in. ILU(0) is a typical choice to precondition iterative solvers and relies only on the levels of fill-in, e.g. sparsity patterns (Saad, 2003). One can obtain better approximation with ILU by using incomplete factorizations with thresholding.

One such technique is the ILU factorization with thresholding (ILUt($p$)). The parameter $p$ defines the number of additional non-zeros allowed per column in the resulting factorization. For the ILUt($p$) decomposition, the algorithm is more complex and involves both dropping values by some prede-fined threshold and controlling the number of possible non-zero values in the factorization. In the case of ILUt($p$), the value $p$ represents additional non-zero values allowed in the factorization per row. The thresholding algorithm provides a more flexible and effective way to approximate the in-verse of a matrix, especially for realistic problems where the numerical values of the matrix elements are important.

The complexity of solving sparse linear systems with matrices in the form of the Choletsky decom-position defined by the number of non-zero elements $\mathcal{O}(\text{nnz})$. This value also defines the storage complexity and the complexity of preconditioner construction.

## A.8 DETAILS ON CORRECTION COEFFICIENT

Table 13: Values of the learned coefficient $\alpha$ from (6) with loss (3).

| Equation | Variance | Grid $32 \times 32$ | | Grid $64 \times 64$ | | Grid $128 \times 128$ | |
| | | IC(0) | ICt(1) | IC(0) | ICt(1) | IC(0) | ICt(1) |
| --- | --- | --- | --- | --- | --- | --- | --- |
| Poisson | — | $-0.159$ | $-0.091$ | $-0.188$ | $-0.095$ | $-0.202$ | $-0.116$ |
| Diffusion | 0.1 | $-0.107$ | $-0.073$ | $-0.099$ | $-0.082$ | $-0.115$ | $-0.086$ |
| Diffusion | 0.5 | $-0.068$ | $-0.034$ | $-0.064$ | $-0.051$ | $-0.094$ | $-0.055$ |
| Diffusion | 0.7 | $-0.066$ | $-0.031$ | $-0.070$ | $-0.043$ | $-0.062$ | $-0.031$ |

