# OpenReview forum: "Learning from Linear Algebra: A Graph Neural Network Approach to Preconditioner Design for Conjugate Gradient Solvers"
_ICLR.cc/2025/Conference — Submitted to ICLR 2025_

### Official Review · Reviewer_ZRtj · 2024-10-27

**Soundness:** 2
**Presentation:** 3
**Contribution:** 2
**Rating:** 5
**Confidence:** 3

**Summary:**

introduces a method for improving preconditioners in the Conjugate Gradient (CG) method using Graph Neural Networks (GNNs). The key contribution is a new scheme, called PreCorrector, which trains a GNN to modify and improve classical preconditioners like incomplete Cholesky (IC). The paper demonstrates that this approach outperforms traditional preconditioning techniques and previously proposed neural network-based preconditioners.

**Strengths:**

1. The method generalizes well to unseen linear systems, indicating robust applicability.
2. The integration of GNNs  seems reasonable.

**Weaknesses:**

1. The novelty of this work appears to be limited, as it closely resembles existing methods. Moreover, there is a lack of thorough discussion and comparison with relevant prior works, such as [Azulay, Yael, and Eran Treister. "Multigrid-augmented deep learning preconditioners for the Helmholtz equation"], [Li, Kangan, Sabit Mahmood Khan, and Yashar Mehmani. "Machine learning for preconditioning elliptic equations in porous microstructures: A path to error control"], and [Li, Kangan, Sabit Mahmood Khan, and Yashar Mehmani. "Machine learning for preconditioning elliptic equations in porous microstructures: A path to error control"]. A more detailed comparison would help to better contextualize the contribution of this work within the existing literature.

2. The problems for validation are relatively esay and small, which are Poisson 2D or diffusion equation.

**Questions:**

1. Could the authors provide more specific details on how the computation time was measured, including the tools or setup used for timing and any relevant conditions that might affect the results?

2. The idea presented seems similar to the method proposed by [Li, Yichen, et al. "Learning preconditioners for conjugate gradient PDE solvers." International Conference on Machine Learning. PMLR, 2023]. However, their method failed with cases involving varying coefficients, as seen in Tables 4 and 5. Could you provide a more detailed explanation of why their approach might have failed, considering differences in the architecture, loss function, and training method? The current explanation in the paragraph "Experiments with neural preconditioner design" is not clear, and it is difficult to understand what sets PreCorrector apart and how it addresses these challenges.

3. Any explanation why the factor $L$ depends on $b$ from L(θ) = GNN(θ, A, b)? Does it mean that in the solving phase, the factor $L$ would vary across iterations in the sense that L(θ) = GNN(θ, A, r), where r is the residual.

---

> ### Author Response · Authors · 2024-11-22
>
> Dear Reviewer ZRtj,
>
> Thank you for your work in reviewing our manuscript. Let us answer the questions from your review.
>
> > Citations
>
> Thank you for sending us relevant papers, we have mentioned them in the "Related work" section in lines 98-101 in the revised manuscript. Although we have limited the number of papers, we would like to clarify the comparison with these papers here.
>
> Both approaches use convolutional neural networks (CNNs). In preconditioner design, CNNs can suffer from the curse of dimensionality, as convolutions scale poorly with matrix growth, since sparse matrices must be materialized as dense ones. Furthermore, message-passing GNNs can be seen as a generalization of convolutional neural networks, which can operate not only on a rectangular grid with a fixed number of neighbours, but also on an arbitrary grid. Moreover, both works are essentially quite different from our approach: these papers propose hybrid preconditioners with neural networks that also perform inference at each step of the iterative solvers. This is very different from the PreCorrector, which is not a preconditioner itself, but is used to create a classical preconditioner from the matrix.
>
> > The novelty of this work appears to be limited, as it closely resembles existing methods.
>
> Although we have not invented the use of GNNs on sparse linear systems, the PreCorrector is, to our knowledge, the first to achieve a better effect on the spectrum than classical preconditioners of the ILU family. Moreover, in our experiments, different realizations of the message passing architecture, node/edge updates, etc. did not change the convergence or the resulting preconditioner quality. We observe that the crucial part of a good neural preconditioner is initialization and stable learning, which is achieved by the PreCorrector architecture.
>
> The GNNs from [Li et al.] have major limitations that limit the quality of the resulting preconditioner: (i) convergence to local minima and (ii) unstable learning. Both are addressed by PreCorrector.
>
> > Could the authors provide more specific details on how the computation time was measured, including the tools or setup used for timing
> and any relevant conditions that might affect the results?
>
> Please see lines 312-316 for details of the comparison setup. Note that while the GPU was used for preconditioner construction with PreCorrector and GNN from [Li et al.], the CPU was used for the classical IC algorithm. Classical IC algorithm is fully sequential and cannot significantly benefit from data parallelization. The $\texttt{ilupp}$ library is a Python API for C++ code, and our implementations of PreCorrector and GNN from [Li et al.] are coded in $\texttt{jax}$, which is JIT-compiled for time computation. For CG, the $\texttt{scipy.sparse.linalg.cg}$ implementation and the $\texttt{scipy.sparse.linalg.LinearOperator}$ class from it are used for time-to-solution measurements.
>
> > Could you provide a more detailed explanation of why their approach might have failed, considering differences in the architecture, loss function, and training method?
>
> The main limitations of the previous work by [Li et al.] are summarised in lines 194-198. Let us discuss them here in greater details. While GNN of [Li et al.] indeed converges and provides a neural IC that can reduce the number of iterations in CG for certain problems, it has a worse effect on the spectrum than classical IC from linear algebra. First, we believe that this is due to convergence to suboptimal local minima, which can be overcome by the starting point in training. In PreCorrector we get such a good initial guess with a classical IC decomposition. Second, the training GNN of [Li et al.] is unstable, since at the very beginning of training one has to compute the loss (5) with a random matrix L. We observed a loss overflow as the matrix size grows. Finally, we observe that we cannot predict the resulting quality of the GNN from [Li et al.]. In our experiments, we were able to compute the condition number for small linear systems: decreasing the loss did not correspond to decreasing the condition number of $P(\theta)^{-1}A$. Thus, the stopping criteria of GNN from [Li et al.] training has to rely on the condition number of the preconditioned system, which is extremely costly. The learning of the PreCorrector starts with the gradient of the pure classical IC (since $\alpha$ is initialized with 0), which ensures stable learning.
>
> Thus, the limitations of GNN from [Li et al.] are (i) convergence to local minima and (ii) unstable learning. Both are fixed by PreCorrector.

---

> ### Author Response · Authors · 2024-11-22
>
> > Any explanation why the factor L depends on b from $L(\theta) = \text{GNN}(\theta, A, b)$?
>
> Input for the GNN is a graph combined out of linear system $Ax = b$, where edges are elements of the matrix and vertices are the rhs $b$ (for Precorrector instead of matrix $A$ we pass $L$ from classical linear algebra decomposition). Note that GNN is only inferenced once before the CG iterations to combine the preconditioner. During the CG iterations, the preconditioners obtained with GNN or PreCorrector are used as usual IC preconditioners from linear algebra. It is therefore not necessary to call PreCorrector for each CG iteration.

---

> > ### Comment · Reviewer_ZRtj · 2024-11-26
> >
> > Thank authors for addressing my questions. I still do not understand why the preconditioner or the PreCorrector should depends on b.
> >
> > Overall, the problems concerned in the current paper are too simple to draw a meaningful conclusion.

---

> > > ### Author Response · Authors · 2024-11-27
> > >
> > > Thank you for pointing out the right-hand side as an input to the GNN. Our work was inspired by the previous paper (Li et al. 2023) where the GNN takes both $A$ and $b$ as inputs. In fact, it is a bug and for a proper preconditioner construction routine, the GNN should not be dependent on $b$. We repeated our experiments with dummies for $b$ in the GNN input (vector of ones for each sample) and the results were exactly the same. So the experiments in the submitted manuscript are valid for a GNN that does not depend on $b$.
> > >
> > > Regarding the test problems, we tested our approach on the datasets that are more complex than the datasets of the previous work (with contrast coefficients). In fact, the previous approach failed on these datasets. We believe that it is possible to create a learnable transformation that will be universal for different sparse matrices to construct the ILU decomposition that will significantly reduce $\kappa(A)$. Considering this learnable transformation as a primary goal, we will create a comprehensive and diverse dataset for it, which will most likely deserve its own research paper.

---

### Official Review · Reviewer_VtSh · 2024-10-27

**Soundness:** 3
**Presentation:** 3
**Contribution:** 3
**Rating:** 6
**Confidence:** 5

**Summary:**

This paper proposes to design CG preconditioners with GNNs. The proposed PreCorrector routine takes legacy preconditioners as input and learns corrections to these legacy preconditioners (e.g., ILU). The proposed method maintains the sparsity pattern of the system matrix and provides better convergence properties in CG iterations. The effectiveness of the proposed method is tested on large-scale diffusion and Poisson's equations. Experimental results show significant speedup over legacy preconditioners.

**Strengths:**

- **Convergence**: The proposed PreCorrector routine does converge faster than traditional methods;
- **Generalization**. The proposed method performs well when dealing with high-contrast coefficients in PDEs. This is considered a challenge to all neural preconditioners/solvers;
- **Sparsity Preservation**. The underlying GNN architecture preserves the sparsity pattern, yielding better performance for succeeding GEMV operations;
- **Experimental Results**. The experiments are comprehensive and the results are convincing, showing significant speedup.

**Weaknesses:**

- **Theoretical study of the loss function**, as explained by the authors themselves;
- **Generalization**. The proposed methods are tested over diffusion and Poisson's equations, which are symmetric positive definite. This is acceptable provided that the CG iterator itself requires this property, yet this limits the potential applications of the proposed method.
- **Limited Comparison**. The paper compares the PreCorrector against classical methods and (Li et al, 2023). The scope is somewhat limited.

**Questions:**

- Have the authors tried to extend the method to non-symmetric indefinite systems? BiCGSTAB is already available for these systems and how does PreCorrector work with BiCGSTAB?
- How does PreCorrector perform when preconditioning different sparsity patterns beyond C(0) v.s. ICt(5)?
- Ablation on the $\alpha$ parameter?
- What specific steps or techniques were used to stabilize the training process of PreCorrector?
- Computational Trade-off: For large systems, does the time saved by reduced CG iterations outweigh the precomputation and GNN inference time?

---

> ### Author Response · Authors · 2024-11-22
>
> Dear Reviewer VtSh,
>
> Thank you for your work in reviewing our manuscript. Let us answer the questions from your review.
>
> > Have the authors tried to extend the method to non-symmetric indefinite systems? BiCSTAB is already available for these systems and how does PreCorrector work with BiCGSTAB?
>
> While we have focused on the linear systems with SPD matrices, the proposed architecture can be generalised to general patterns: one should use ILU instead of IC and the GNN neural network should predict the whole graph, not only the one corresponding to the lower triangular part of the matrix (to obtain the factors $L$ and $U$). However, different types of matrices besides SPD and a larger number of iterative solvers to solve systems with them deserve their own research paper.
>
> > How does PreCorrector perform when preconditioning different sparsity patterns beyond IC(0) v.s. ICt(5)?
>
> If we understand your question correctly, you asked what sparsity patterns can be used with PreCorrector. Actually, the sparsity pattern for ICt(k) decomposition can vary depending on the chosen threshold for each new matrix. Also, more advanced IC algorithms (e.g. with pivoting) can be used to get even better preconditioners with probably more complex patterns. Overall, with PreCorrector we do not need to define the sparsity pattern ourselves and hope that it will work well. We rely on classical algorithms to get the pattern from the classical decomposition, which is already good.
>
> We also tried to artificially increase the sparsity pattern of initial $A$ for GNN from [Li et al.] by padding to larger pattern $k$ in IC($k$) (e.g. IC($1$), IC($2$)). The results were almost identical to those obtained by just passing $A$.
>
> > Ablation on the \alpha parameter?
>
> The $\alpha$ parameter is part of the neural network weights learned during training. Therefore there is no ablation study for $\alpha$. It is worth noting that $\alpha$ is equal to $0$ at the very beginning of training, which provides first gradients from pure classical IC decomposition and ensures stable training in the very first step.
>
> > What specific steps or techniques were used to stabilize the training process of PreCorrector?
>
> The PreCorrector itself is built to stabilise the training process by (i) starting with a good initial guess as in classical IC and (ii) ensuring a valid gradient in the very first step starting with $\alpha = 0$. In addition, we normalize the matrix by its largest element by absolute value before inputting it to GNN.
>
> > Computational Trade-off: For large systems, does the time saved by reduced CG iterations outweigh the precomputation and GNN inference time?
>
> Speaking of inference time, for larger systems the PreCorrector call becomes cheaper and thus the effect of the number of CG iterations is less affected by the construction overhead (Tables 1-3). Please note that PreCorrector is not a preconditioner itself, it is used to create one. When preconditioners are combined with PreCorrector, there is no difference in usage during CG compared to preconditioners designed using classical methods. Moreover, it has exactly the same algorithmic complexity as the classical preconditioner it is made of (e.g. PreCorrector[IC($0$)] and IC($0$)). See also Appendix A.5 for more details on scalability.
>
> The impact of PreCorrector’s training overhead is difficult to assess, as it directly depends on the problem you are trying to solve and the degree of generalization required. In general, any data-driven approach requires additional overhead to train the model. On the other hand, some applications require solving linear systems with the same matrix multiple times. For these settings, the ability of PreCorrectors to generalise (Figure 4) makes the training overhead less significant.

---

> ### Comment · Reviewer_VtSh · 2024-11-26
>
> Thank you for your clarification. I am keeping my original rating.

---

### Official Review · Reviewer_X4c7 · 2024-11-01

**Soundness:** 3
**Presentation:** 3
**Contribution:** 2
**Rating:** 5
**Confidence:** 5

**Summary:**

This paper proposes an approach to computing a preconditioner for the conjugate gradient method for solving large linear systems. The approach improves Li et al. (2023) in that rather than learning the nonzeros of the incomplete Cholesky factor, it learns a correction to the factor. The authors show that their approach can outperform the standard incomplete Cholesky (IC) preconditioner, addressing a drawback of Li et al. (2023), which is less performant than IC.

**Strengths:**

- The idea of learning a correction rather than learning the nonzeros of a preconditioner is novel.

- The learned preconditioner can generalize to different grid sizes and different parameters of the PDE.

- The proposed approach can outperform the standard incomplete Cholesky (IC) preconditioner, addressing a drawback of Li et al. (2023), which is less performant than IC.

**Weaknesses:**

- The paper is incremental in that a majority of the basis of the work comes from Li et al. (2023).

- The work stresses the use of a different loss function (Eqn (3) rather than Eqn (2)), but the argument is dubious. The authors reformulate Eqn (3) to Eqn (5) in practice, which, however, can also be obtained directly from Eqn (2). Hence, the authors' argument that they use a better loss function ((3) against (2)) does not hold. For more details, see a related question in the following "Questions" section.

**Questions:**

- While the authors have made clear how the training loss (3) is reformulated to (5), one should note that (2) can also be reformulated to (5), because $b_i = A x_i$. The subtlety lies in whether $x_i$ or $b_i$ is drawn from the standard normal. When one starts with the loss (3), $b_i$ is drawn from the standard normal, which causes the trouble of computing $x_i$ (which requires solving linear systems). In contrast, when one starts with the loss (2), one may draw $x_i$ from the standard normal, which causes no difficulty in computing $b_i$. In this regard, the loss (2) is even better than (3) in practice.

- How does training time compare with solution time? Does the Pre-time column in Tables 1 to 4 mean the evaluation of the GNN or the training of the GNN?

---

> ### Author Response · Authors · 2024-11-22
>
> Dear Reviewer X4c7,
>
> Thank you for your work in reviewing our manuscript. Let us answer the questions from your review.
>
> > The paper is incremental in that a majority of the basis work comes from Li et al. (2023)
>
> Although we have not invented the use of GNNs on sparse linear systems, the PreCorrector is, to our knowledge, the first to achieve a better effect on the spectrum than classical preconditioners of the ILU family. Moreover, in our experiments, different realizations of the message passing architecture, node/edge updates, etc. did not change the convergence or the resulting preconditioner quality. We observe that the crucial part of a good neural preconditioner is initialization and stable learning, which is achieved by the PreCorrector architecture.
>
> The GNNs from [Li et al.] have major limitations that limit the quality of the resulting preconditioner: (i) convergence to local minima and (ii) unstable learning. Both are addressed by PreCorrector.
>
> > While the authors have made clear how the training loss (3) is reformulated to (5), one should note that (2) can also be reformulated to (5).
>
> Indeed, the Hutchinson trick can be applied to loss (2). In addition, we used the Hutchinson trick for training with loss (2) (please note the caption for Table 6). The main idea is that using loss (3) is more beneficial than using loss (2) because loss (3) emphasises the low frequencies where CG has the most problems. While training with loss (2) can be done in an unsupervised manner, the resulting preconditioner will have a worse effect on the spectrum than when training with loss (3) (see Table 6). Moreover, when training with loss (2), the resulting preconditioner will be worse than the classical IC preconditioner in terms of CG iterations (see number of iterations "Loss (2)" from Table 6 and "IC(0)" for grid 64x64 in Table 1). Thus, loss (5) is only needed to avoid explicit inverse calculation in loss (3).
>
> > How does training time compare with solution time? Does the Pre-time column in Tables 1 to 4 mean the evaluation of the GNN or the training of the GNN?
>
> Pre-time means the construction time of the neural and classical preconditioners. Note that the pre-time for the PreCorrector is calculated including both the inference of the GNN and the construction of the classical preconditioner.
>
> Training PreCorrector on the most complex linear system (grid 128, variance 0.7) converged in about 500 epochs or 170 minutes (see lines 306-316 for experiments environment info). Unfortunately, any data-driven approach requires additional overhead to train the model. On the other hand, some applications require solving linear systems with the same matrix several times. For these settings, the ability of PreCorrector to generalize (see Figure 4) makes it a more favourable approach.

---

> > ### Comment · Reviewer_X4c7 · 2024-11-22
> > **One question unanswered**
> >
> > Thank you for the rebuttal. One question for which I am still waiting for an answer is, in your loss (5) reformulated from the advocated loss (3), how is $x_i$ generated? It appears that you still need to solve the linear system $A x_i = b_i$ given a sampled $b_i$ (which is the original problem being tackled).

---

> > > ### Author Response · Authors · 2024-11-23
> > >
> > > For training with loss (3), reformulated as loss (5), one needs a dataset of $N$ samples of the linear system $A_i x_i = b_i$, where $x_i$ is obtained by solving the linear system. However, in practice this should not be an additional problem, since the solution is not needed during inference, but is part of the dataset preparation for training.
> > >
> > > Note that training with loss (2) can indeed be done unsupervised without solving $A_i x_i = b_i$. However, since models with loss (2) cannot outperform preconditioners with classical algorithms for IC (see number of iterations "Loss (2)" from Table 6 and "IC(0)" for grid 64x64 in Table 1), in practice neural network models trained with loss (2) will not be applicable, since it is better just to use classical IC (a very simple and efficient algorithm) without any training at all.

---

> > > > ### Comment · Reviewer_X4c7 · 2024-11-24
> > > >
> > > > Thank you for supplementing additional information. I can see that (3) is better than (2). The need to solve $A_ix_i=b_i$ to create a dataset, however, is a computational concern, especially when this dataset is large.

---

### Official Review · Reviewer_7yrG · 2024-11-03

**Soundness:** 2
**Presentation:** 3
**Contribution:** 2
**Rating:** 3
**Confidence:** 4

**Summary:**

The paper titled "Learning from Linear Algebra: A Graph Neural Network Approach to Preconditioner Design for Conjugate Gradient Solvers" presents a novel method for designing preconditioners using graph neural networks (GNNs). It aims to improve the efficiency of solving large linear systems that arise in computational science and engineering, particularly those characterized by parametric partial differential equations. The authors argue that their approach, termed "PreCorrector," leverages existing linear algebra techniques and demonstrates superior performance in numerical experiments compared to traditional and other neural network-based methods.

**Strengths:**

- **Innovative Approach**: The use of GNNs to enhance traditional preconditioners is a fresh perspective in the domain of numerical linear algebra.
- **Theoretical Foundation**: The paper provides a strong theoretical basis for the loss function and discusses its implications on the preconditioner's performance.
- **Numerical Validation**: Extensive experiments validate the proposed method, showcasing its effectiveness in reducing the condition number of systems, which is crucial for the performance of iterative solvers.

**Weaknesses:**

- **Limited Scope**: While the paper presents promising results, the experiments are confined to a specific class of parametric PDEs. The generalizability of the results to broader contexts remains unclear.
- **Comparison with State-of-the-Art**: The comparison with existing methods could be more comprehensive. Many contemporary techniques in preconditioner design and GNN applications are not adequately addressed.
- **Complexity of Implementation**: The proposed method may introduce additional complexity in practical implementations, which could deter its adoption in industry settings.

**Questions:**

1. Can the authors provide more insights into how the proposed GNN architecture can be adapted or generalized to other types of linear systems?
2. How do the computational costs of training and implementing the GNN compare to those of traditional preconditioner design methods?
3. Are there plans to evaluate the proposed method on a wider array of problems, particularly those encountered in real-world engineering applications?

---

> ### Author Response · Authors · 2024-11-22
>
> Dear Reviewer 7yrG,
> Thank you for your work in reviewing our manuscript. Let us answer the questions from your review.
>
> > The comparison with existing methods could be more comprehensive. Many contemporary techniques in preconditioner design and GNN applications are not adequately addressed.
>
> Our main goal is to create a better preconditioner than the classic ones in the ILU family. So the comparison is made with that in mind. Another reviewer shared two relevant papers, which we have mentioned in the "Related work" paragprah in lines 98-101 in the revised manuscript. Although we have limited the number of pages, we would like to clarify the comparison with these papers here.
>
> Both approaches use convolutional neural networks (CNNs). In preconditioner design, CNNs can suffer from the curse of dimensionality, as convolutions scale poorly with matrix growth, since sparse matrices must be materialised as dense ones. Furthermore, message-passing GNNs can be seen as a generalization of convolutional neural networks, which can operate not only on a rectangular grid with a fixed number of neighbours, but also on an arbitrary grid. Moreover, both works are essentially quite different from our approach: these papers propose hybrid preconditioners with neural networks that also perform inference at each step of the iterative solvers. This is very different from the PreCorrector, which is not a preconditioner itself, but is used to create a classical preconditioner from the matrix.
>
> > The proposed method may introduce additional complexity in practical implementations, which could deter its adoption in industry settings.
>
> The proposed approach is a very shallow neural network (2754 parameters), so the inference of such a network with a sparse linear system is negligible compared to the iteration speed-up, which is also illustrated in our experiments. Furthermore, for real industrial applications, the C++ and/or CUDA implementation will be used. These will further reduce the computation time. See also Appendix A.5 for details on scalability.
>
> On the other hand, some applications require solving linear systems with the same matrix several times. For these settings, the ability of PreCorrector to generalize (see Figure 4) makes it a more favourable approach.
>
> > Can the authors provide more insights into how the proposed GNN architecture can be adapted or generalized to other types of linear systems?
>
> While we have focused on the linear systems with SPD matrices, the proposed architecture can be generalized to general patterns: one should use ILU instead of IC and the GNN neural network should predict the whole graph, not only the one corresponding to the lower triangular part of the matrix (to obtain the factors $L$ and $U$). However, different types of matrices besides SPD and a larger number of iterative solvers to solve systems with them deserve their own research paper.
>
> > How do the computational costs of training and implementing the GNN compare to those of traditional preconditioner design methods?
>
> The inference (preconditioner construction) time with the proposed approach is reported as "Pre-time" in the Tables 1-4 and Tables 7-10 (for PreCorrector, this time includes both GNN inference and classical preconditioner construction time). As the matrix size increases, the construction time of both classical and neural preconditioners becomes less significant. When the preconditioner is combined with the PreCorrector, there is no difference in its use during CG compared to preconditioners designed using classical methods. Moreover, it has exactly the same algorithmic complexity as the classical preconditioner it is composed of (e.g. PreCorrector[IC(0)] and IC(0)).
>
> Training PreCorrector on the most complex linear system (grid 128, variance 0.7) converged in about 500 epochs or 170 minutes (see lines 306-316 for experiments environment info). Unfortunately, any data-driven approach requires additional overhead to train the model.
>
> Speaking about the implementation of PreCorrect, it is basically a combination of well-known operations: classical algorithm for IC decomposition, multilayer perceptrons for encoders, decoders and update functions in GNN, and message-passing architecture, which mostly consists of common graph operations (e.g. collecting information about node neighbourhood).
>
> > Are there plans to evaluate the proposed method on a wider array of problems, particularly those encountered in real-world engineering applications?
>
> Yes, we have plans to evaluate the PreCorrector on the real problems of various applications. The application of PreCoorector is our next primary goal.

---

### Meta-Review · Area_Chair_WavS · 2024-12-21

**Metareview:**

Here the authors propose a method for designing preconditioners based on training a GNN. While some of the reviewers point out the novelty of using GNNs for learning preconditioners, the reviewers also note weaknesses in the scope and applicability of the work beyond the specific problems presented in the paper along with questions regarding the overall computation cost of training a GNN to produce a preconditioner, particularly for large datasets.

Overall, the consensus of the reviewers is to reject the paper, and I would encourage the authors to incorporate the feedback of the reviewers in preparing future versions of the manuscript.

**Additional Comments On Reviewer Discussion:**

The authors were active in responding to the questions from the reviewers and engaging with the reviewers in further follow-up questions.

---

### Decision · Program_Chairs · 2025-01-22

Reject